# MODEL-BASED ROBUST DEEP LEARNING: GENERALIZING TO NATURAL, OUT-OF-DISTRIBUTION DATA

## ABSTRACT

While deep learning (DL) has resulted in major breakthroughs in many applications, the frameworks commonly used in DL remain fragile to seemingly innocuous changes in the data. In response, adversarial training has emerged as a principled approach for improving the robustness of DL against norm-bounded perturbations. Despite this progress, DL is also known to be fragile to unbounded shifts in the data distribution due to many forms of natural variation, including changes in weather or lighting in images. However, there are remarkably few techniques that can address robustness to natural, out-of-distribution shifts in the data distribution in a general context. To address this gap, we propose a paradigm shift from perturbation-based adversarial robustness to *model-based robust deep learning*. Critical to our paradigm is to obtain *models of natural variation*, which vary data over a range of natural conditions. Then by exploiting these models, we develop three novel model-based robust training algorithms that improve the robustness of DL with respect to natural variation. Our extensive experiments show that across a variety of natural conditions in twelve distinct datasets, classifiers trained with our algorithms significantly outperform classifiers trained via ERM, adversarial training, and domain adaptation techniques. Specifically, when training on ImageNet and testing on various subsets of ImageNet-c, our algorithms improve over baseline methods by up to 30 percentage points in top-1 accuracy. Further, we show that our methods provide robustness (1) against natural, out-of-distribution data, (2) against multiple simultaneous distributional shifts, and (3) to domains entirely unseen during training.

## 1 INTRODUCTION

The last decade has seen remarkable progress in deep learning (DL), which has prompted wide-scale integration of DL frameworks into myriad application domains (LeCun et al., 2015). In many of these applications, and in particular in *safety-critical* domains, it is essential that the DL systems are robust and trustworthy (Dreossi et al., 2019). However, it is now well-known that DL is fragile to seemingly innocuous changes to the input data (Szegedy et al., 2013). Indeed, well-documented examples of fragility to carefully-designed noise can be found in a variety of contexts, including image classification (Madry et al., 2017), clinical trials (Papangelou et al., 2018), and robotics (Melis et al., 2017). Accordingly, a number of *adversarial training* algorithms (Goodfellow et al., 2014b; Wong & Kolter, 2017) as well as certifiable defenses (Raghunathan et al., 2018; Fazlyab et al., 2019a) have recently been proposed, which have provided a rigorous framework for improving the robustness of DL against norm-bounded perturbations (Fazlyab et al., 2019b; Dobriban et al., 2020).

Despite this encouraging progress, very recent papers have unanimously shown that DL is also fragile to unbounded shifts in the data-distribution due to a wide range of natural phenomena (Djolonga et al., 2020; Taori et al., 2020; Hendrycks et al., 2020; Hendrycks & Dietterich, 2019). For example, in image classification, such shifts include changes due to lighting, blurring, or weather conditions (Pei et al., 2017; Chernikova et al., 2019). However, there are remarkably few general, principled techniques that have been shown to provide robustness against these forms of out-of-distribution, natural variation (Hendrycks et al., 2019a). Furthermore, as these unseen distributional shifts are arguably more common in safety-critical domains, the task of designing algorithms that generalize to natural, out-of-distribution data is an important and novel challenge for the DL community.

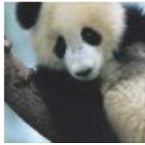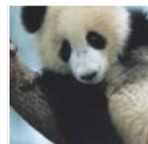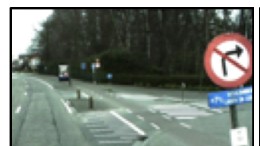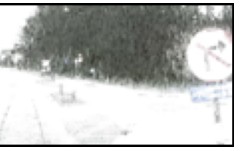

(a) **Perturbation-based adversarial example.** In a perturbation-based robustness setting, an input datum (left) is perceptually indistinguishable from a corresponding adversarial example (right).

(b) **Natural variation.** In this paper, we study robustness with respect to natural variation. For example, differences in weather conditions such as snow illustrate one form of natural variation.

Figure 1: **A new notion of robustness.** Past work has focused on perturbation-based adversarial examples, such as Figure 1a. In this paper, we focus on robustness with respect to natural variation, shown in Figure 1b, which often does not obey perceptual or norm-bounded constraints.

In this paper, we propose a paradigm shift from perturbation-based adversarial robustness to *model-based robust deep learning*. Our goal is to provide principled, general algorithms that can be used to train neural networks to be robust against natural, out-of-distribution shifts in data. Our experiments show that across a variety of challenging, naturally-occurring conditions, such as variation in lighting, haze, rain, and snow, and across various datasets, including SVHN, GTSRB, CURE-TSR, and ImageNet, classifiers trained with our model-based algorithms significantly outperform standard DL baselines, adversarially-trained classifiers, and, when applicable, domain adaptation methods.

**Contributions.** The contributions of this paper can be summarized as follows:
- **Paradigm shift.** We propose a paradigm shift from perturbation-based robustness to model-based robust deep learning, where models of natural variation express changes due to natural conditions.
- **Optimization-based formulation.** We formulate a novel model-based robust training problem by constructing a general robust optimization procedure to search for challenging natural variation.
- **Models of natural variation.** For many challenging forms of natural variation, we use deep generative models to learn models of natural variation that are consistent with realistic conditions.
- **Novel algorithms.** We propose a family of novel robust training algorithms that exploit models of natural variation to improve the robustness of DL against worst-case natural variation.
- **Out-of-distribution robustness.** We show that our algorithms are the first to consistently provide robustness against natural, out-of-distribution shifts that frequently occur in real-world environments, including snow, rain, fog, and brightness on SVHN, GTSRB, CURE-TSR, and ImageNet.
- **ImageNet-c robustness.** We show that our algorithms can significantly improve the robustness of classifiers trained on ImageNet and tested on ImageNet-c by as much as 30 percentage points.
- **Robustness to simultaneous distributional shifts.** We show that our methods are composable and can improve robustness to multiple simultaneous sources of natural variation. To evaluate this feature, we curate several new datasets, each of which has two simultaneous distributional shifts.
- **Robustness to unseen domains.** We show that models of natural variation can be reused on datasets that are entirely unseen during training to improve out-of-distribution generalization.

## 2 PERTURBATION-BASED ROBUSTNESS: APPROACHES AND LIMITATIONS

In this paper, we consider a standard classification task in which training data $(x, y) \sim \mathcal{D}$ is distributed according to a joint distribution $\mathcal{D}$ over instances $x \in \mathbb{R}^d$ and labels $y \in [k] := \{0, 1, \ldots, k\}$. In this setting, given a finite training sample drawn i.i.d. from $\mathcal{D}$, the goal of the learning problem is to obtain a classifier $f_w$ parameterized by weights $w \in \mathbb{R}^p$ such that $f_w$ can correctly predict the labels $y$ corresponding to new instances $x$ drawn i.i.d. from $\mathcal{D}$. In practice, one can learn $f_w$ by approximately solving the non-convex empirical risk-minimization (ERM) problem $\arg\min_w \mathbb{E}[\ell(x, y; w)]$ where $\ell$ is a suitable loss-function. However, neural networks trained using ERM are known to be susceptible to *adversarial attacks*. This means that given a datum $x$ with a corresponding label $y$, one can find another datum $x^{\text{adv}}$ such that $x$ is close to $x^{\text{adv}}$ in a given Euclidean norm and $x^{\text{adv}}$ is predicted by the learned classifier as belonging to a different class $c \neq y$. If such a datum $x^{\text{adv}}$ exists, it is called an *adversarial example*. This is illustrated in Figure 1a; although these pandas look identical, they were classified differently in (Goodfellow et al., 2014b).

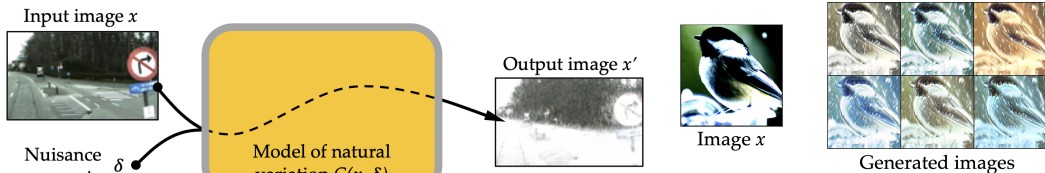

(a) Models take the form $G(x, \delta)$, where $\delta$ is a *nuisance parameter* that describes how the output image $x' := G(x, \delta)$ is varied.

(b) Input image $x$ and corresponding generated images for a learned model of natural variation on ImageNet.

Figure 2: In this paper, we introduce *models of natural variation* to describe natural transformations.

The dominant paradigm toward improving robustness against adversarial examples relies on a robust optimization perspective wherein neural networks are trained to correctly classify worst-case perturbations of data (Madry et al., 2017; Wong & Kolter, 2017). This can be formulated as follows:

$$\arg \min_{w} \mathbb{E}_{(x,y) \sim \mathcal{D}} \left[ \max_{\delta \in \Delta} \ell(x + \delta, y; w) \right] \tag{1}$$

We can think of (1) as comprising two coupled optimization problems: an inner maximization problem in which we seek a challenging perturbation and an outer minimization problem in which we seek weights that lead to strong classification performance.

**Limitations of perturbation-based robustness.** Despite remarkable progress toward improving the robustness of DL against norm-bounded perturbations, there are significant limitations to adversarial training. Notably, DL is known to be fragile to many forms of *natural variation*, which cannot be described by small perturbations $x \mapsto x + \delta$. In image classification, such natural variation includes changes in weather or background color (Eykholt et al., 2018; Hendrycks et al., 2019b; Hosseini & Poovendran, 2018), spatial transformations such as rotation or scaling (Xiao et al., 2018b; Kari-anakis et al., 2016), and sensor-based attacks (Kurakin et al., 2016). Because such transformations frequently arise in safety-critical domains, it is critically important for the DL community to develop algorithms that are robust against out-of-distribution, natural variation in data. In this paper, we specifically address this challenge by proposing a principled, optimization-based framework which can be used in general settings to provide robustness against arbitrary sources of natural variation.

## 3  A NEW ROBUSTNESS PARADIGM: MODEL-BASED ROBUST DEEP LEARNING

Underlying the task of improving the robustness of neural networks against natural, out-of-distribution data are two fundamental challenges. Firstly, unlike in the adversarial robustness community, in real-world, safety-critical environments, data can vary in unknown and highly nonlinear ways. Thus, the first step toward building a robust training procedure must be to design a mechanism that accurately describes how data varies in such environments. Next, assuming a suitable model of natural variation, the second challenge is to formulate a training procedure that exploits this model toward improving robustness. In this section, we present novel solutions to each of these challenges.

### 3.1  MODELS OF NATURAL VARIATION

In order to effectively model sources of natural variation in a domain-agnostic setting, we will abstractly define *models of natural variation*. Concretely, a model of natural variation $G(x, \delta)$ is a map that describes how an input datum $x$ can be naturally varied by a *nuisance parameter* $\delta$ resulting in a new datum $x' := G(x, \delta)$. Ideally, for a fixed datum $x$, varying the nuisance parameter $\delta$ should vary the severity of the natural conditions in the generated datum $x'$. An example of such a model is shown in Figure 2, where an image $x$ on the left (in this case, in sunny weather) can be naturally varied by $\delta$ and consequently transformed into the image $x'$ on the right (in snowy weather). In the remainder of this subsection, we consider cases in which (1) a model $G$ is known a priori, and (2) a model $G$ is unknown and therefore must be learned offline from data. In this second case in which models of natural variation must be learned, we propose a method for obtaining these models.

**Known models of natural variation.** In many problems, a model $G(x, \delta)$ is known a priori due to intrinsic geometric structure. For example, there is underlying structure that describes how data can

be rotated, translated, or scaled; models for rotating an image can be characterized by $G(x, \delta) = R(\delta)x$ where $R(\delta)$ is a rotation matrix and $\delta \in \Delta := [0, 2\pi)$. In prior work, this idea has been used to train classifiers to be robust to rotation and scaling (Engstrom et al., 2017; Kamath et al., 2020).

**Learning models of natural variation from data.** In many situations, models natural variation are not known a priori or are too costly to obtain. For example, consider Figure 2 in which a model $G(x, \delta)$ takes an image $x$ of a street sign in sunny weather and maps it to an image $x' := G(x, \delta)$ in snowy weather. Even though there is a relationship between the two images, obtaining a model $G$ relating these two domains is extremely challenging if we resort to geometric structure. For such problems we advocate for *learning* the model $G$ from data. To do so, we assume that we have access to two unpaired domains $A$ and $B$ that are drawn from a common distribution. Domain $A$ contains the original data, such as the images with sunny weather, and domain $B$ contains naturally transformed data, such as images with snow. Ideally, a model of natural variation should learn to map images from domain $A$ to corresponding images with different levels of natural variation captured by the images of domain $B$. In our experiments section, we rely on the MUNIT framework (Huang et al., 2018), which combines two autoencoders and two generative adversarial networks (Goodfellow et al., 2014a), to learn mappings between domains $A$ and $B$. Furthermore, we note that many choices unpaired, unconditional image-to-image translation networks satisfy our criteria for $G$, and in future work we plan to investigate the efficacy of these architectures. In Appendix A, we describe parallel experiments that we carried out with two other architectural choices for $G$, and we fully characterize the MUNIT architecture used in our experiments.

### 3.2 MODEL-BASED ROBUST TRAINING FORMULATION

The model-based robust training paradigm that we propose retains the basic elements of adversarial training described in Section 2. Our point of departure from the classical adversarial training formulation is in the choice of the so-called adversarial perturbation. In this paper, we assume that data can be transformed according to a model of natural variation $G(x, \delta)$ by choosing different values of $\delta$ from a given *nuisance space* $\Delta$. The goal of the model-based approach is to train a classifier that achieves high accuracy both on a test set drawn i.i.d. from $\mathcal{D}$ and on *more-challenging* test data that has been subjected to the source of natural variation that $G$ models. This perspective can be captured by the following robust optimization problem:

$$\min_{w} \mathbb{E}_{(x,y) \sim \mathcal{D}} \left[ \max_{\delta \in \Delta} \ell(G(x, \delta), y; w) \right]. \tag{2}$$

In the inner maximization problem of this formulation, given an instance-label pair $(x, y)$, we seek a vector $\delta^* \in \Delta$ that produces a corresponding instance $x' := G(x, \delta^*)$ which gives rise to high loss values $\ell(G(x, \delta^*), y; w)$ under the current weight $w$. One can think of this vector $\delta^*$ as characterizing the *worst-case* nuisance that can be generated by the model $G(x, \delta^*)$ for the original instance $x$. After solving this inner problem, we solve the outer minimization problem by finding weights $w$ that minimize the risk against the challenging instance $G(x, \delta^*)$. By training the network to correctly classify this worst-case data, the goal is to become invariant to the model $G(x, \delta)$ for any $\delta \in \Delta$.

## 4 MODEL-BASED TRAINING ALGORITHMS

We now assume that we have access to a suitable model of natural variation $G(x, \delta)$ and shift our attention toward exploiting $G$ in the development of novel robust training algorithms. In the empirical version of (2), rather than assuming access to the full joint distribution $\mathcal{D}$, we assume that we are given given a set of i.i.d. samples $\mathcal{D}_n := \{(x_j, y_j)\}_{j=1}^{n}$ drawn from $\mathcal{D}$. Thus we have:

$$w^\star \in \arg\min_{w \in \mathbb{R}^p} \sum_{j=1}^{n} \left[ \max_{\delta \in \Delta} \ell\left(G\left(x_j, \delta\right), y_j; w\right) \right]. \tag{3}$$

Note that when $w$ parameterizes a neural network, (3) is a nonconvex-nonconcave min-max problem, which is difficult to solve exactly. Thus, we resort to approximate optimization techniques for solving (3). Specifically, we propose three algorithmic variants: *Model-based Adversarial Training* (MAT), *Model-based Robust Training* (MRT), and *Model-based Data Augmentation* (MDA). Pseudocode for MAT is given in Algorithm 1; pseudocode for MRT and MDA as well as further discussion of these algorithms is given in Appendix B. At a high level, each of these methods alternates between solving the outer minimization problem and the inner maximization problem. To this

---

**Algorithm 1** Model-based Adversarial Training (MAT)

> **Input:** Data sample $\mathcal{D}_n = \{(x_j, y_j)\}_{j=1}^n$, number of steps $k$, step size $\alpha$
> **Output:** Learned weight $w$

1: **repeat**
2:     **for** minibatch $B_m := \{(x_1, y_1), (x_2, y_2), \ldots, (x_m, y_m)\} \subset \mathcal{D}_n$ **do**
3:         Initialize $\delta := (\delta_1, \delta_2, \ldots, \delta_m) \leftarrow (0_q, 0_q, \ldots, 0_q)$
4:         **for** $k$ steps **do**
5:             $g \leftarrow \nabla_\delta \sum_{j=1}^m \ell(G(x_j, \delta_j), y_j; w)$
6:             $\delta \leftarrow \Pi_\Delta[\delta + \alpha g]$        # $\Pi_\Delta$ denotes projection onto the set $\Delta$
7:         **end for**
8:         $g \leftarrow \nabla_w \sum_{j=1}^m [\ell(G(x_j, \delta_j), y_j; w) + \lambda \cdot \ell(x_j, y_j; w)]$
9:         $w \leftarrow \text{Update}(g, w)$       # Update can be SGD, Adam, Adadelta, etc.
10:     **end for**
11: **until** convergence

---

end, each of these algorithms uses SGD to solve the outer problem; the methods differ in how they seek solutions to the inner problem, and in what follows, we describe each of these procedures in more detail. In each algorithm, given a datum $(x, y)$, the solution $\delta^\star$ to the inner problem is used to create a new datum $(G(x, \delta^\star), y)$ that is added to the training set before solving the outer problem.

**Model-based Adversarial Training.** In MAT, we seek an exact solution to the inner problem by performing $k$ steps of gradient ascent in $\delta$ on the objective $\ell(G(x, \delta), y; w)$. The resulting nuisance parameter $\delta^\star$ is one that causes $\ell(G(x, \delta^\star), y; w)$ to have high loss under the current weight $w$.

**Model-based Robust Training.** In MRT, we first randomly sample $\delta_i \in \Delta$ for $i \in [k]$. We then select the $i^\star \in [k]$ such that $\ell(G(x, \delta_{i^\star}), y; w)$ is maximized. In this way, rather than exactly solving the inner problem, MRT uses a sampling-based approach to finding challenging data $G(x, \delta_{i^\star})$.

**Model-based Data Augmentation.** In MDA, rather than explicitly trying to solve the inner problem, we seek a *diversity* of naturally-varying data rather than the "worst-case." In this way, MDA samples $\delta_i \in \Delta$ for $i \in [k]$ and then appends $\{G(x, \delta_i), y\}_{i=1}^k$ to the training dataset.

## 5 EXPERIMENTS

We present experiments in five different settings over twelve distinct datasets to demonstrate the broad applicability of MBRDL. First, in Sections 5.1-5.2, we show that our algorithms are the first to consistently provide out-of-distribution robustness across a range of challenging corruptions, including shifts in brightness, contrast, snow, fog, frost, and haze on CURE-TSR, ImageNet, and ImageNet-c. In Section 5.3, we curate several new datasets containing simultaneous sources of natural variation, and we then show that models of natural variation can be composed to provide robustness against these simultaneous shifts. In Section 5.4, we show that models of natural variation trained on a fixed dataset can be reused to provide robustness on datasets entirely unseen during training. Finally, in Section 5.5, we assume access to unlabeled data corresponding to a fixed domain shift, and we compare our algorithms to suitable baselines, including domain adaptation methods. Throughout these experiments, we use the notation "source $(A{\rightarrow}B)$" to denote a distributional shift from domain $A$ to domain $B$. For example, "contrast (low→high)" will denote a shift from low-contrast to high-contrast. Images from domains $A$ and $B$ for each of the shifts used in this paper are available in Appendix A. We note that our experiments contain domains with both *natural* and *artificially-generated* variation; details concerning how we extracted non-artificial variation can be found in Appendix D. Architecture and hyperparameter details are given in Appendix C.

### 5.1 OUT-OF-DISTRIBUTION ROBUSTNESS

In many applications, one might have data corresponding to low levels of natural variation, such as a dusting of snow in images of street signs. However, it is often difficult to collect data corresponding to high levels of natural variation, such as images taken during a blizzard. In such cases, we show that our algorithms can be used to provide significant out-of-distribution robustness against data with high levels of natural variation by training on data with relatively low levels of the same source of

Table 1: **Out-of-distribution robustness.** In each experiment, we train a model of natural variation to map from challenge-level 0 to challenge-level 2 data from different subsets of CURE-TSR. We then perform model-based training using challenge-level 0 data and test on challenge-levels 3-5.

| CURE-TSR subset | Test accuracy (top-1) on levels 3, 4, and 5 | | | | | | | | |
|---|---|---|---|---|---|---|---|---|---|
| | ERM + Aug | | | PGD + Aug | | | MRT | | |
| | 3 | 4 | 5 | 3 | 4 | 5 | 3 | 4 | 5 |
| Snow | 86.5 | 74.8 | 60.9 | 82.9 | 77.3 | 61.8 | **88.0** | **77.8** | **70.7** |
| Haze | 55.2 | 54.0 | 47.5 | 83.8 | 63.1 | 53.4 | **83.9** | **79.1** | **70.1** |
| Decolorization | 87.9 | 85.1 | 78.8 | 84.7 | 75.2 | 64.9 | **90.5** | **89.6** | **89.4** |
| Rain | 72.7 | 71.7 | 66.9 | 68.9 | 66.4 | 60.5 | **80.7** | **78.7** | **74.8** |

Table 2: **ImageNet to ImageNet-c robustness.** In each experiment, we train a model of natural variation to map from classes 0-9 of ImageNet to the same classes from a subset of ImageNet-c. Next, we use this model to perform model-based training on classes 10-59 of ImageNet, and we test each network on classes 10-59 from the same subset ImageNet-c on which the model was trained.

| Model dataset (classes 0-9) | Training dataset (classes 10-59) | Test dataset (classes 10-59) | Test accuracy (top-1/top-5) | | | | | |
|---|---|---|---|---|---|---|---|---|
| | | | ERM | | AugMix | | MDA | |
| Snow | | Snow | 20.9 | 49.9 | 1.10 | 8.3 | **31.1** | **61.2** |
| Contrast | ImageNet | Contrast | 41.1 | 73.4 | 0.72 | 6.76 | **50.0** | **79.5** |
| Brightness | | Brightness | 26.9 | 59.2 | 0.56 | 5.20 | **53.0** | **81.7** |
| Frost | | Frost | 16.3 | 39.0 | 29.5 | 58.4 | **36.0** | **67.2** |

natural variation. To do so, we use data from the CURE-TSR dataset (Temel et al., 2019), which contains images of street signs divided into subsets according to various sources of natural variation and corresponding severity levels. For example, for images in the "snow" subset, level 0 corresponds to no snow, whereas level 5 corresponds to a full blizzard. Thus, for each row of Table 1, we use unlabeled data from levels 0 and 2 to learn a model of natural variation corresponding to a given source of natural variation in CURE-TSR. We then train classifiers using MDA with labeled level 0 data. We also train classifiers using ERM and PGD using the labeled data from levels 0 and 2. We then test all classifiers on data from levels 3, 4, and 5. Note that while this is an unfair comparison for our methods, given that the model-based algorithms are not given access to labeled level 2 data, our algorithms still outperform the baselines by as much as 20 percentage points on level 5 data.

## 5.2 MODEL-BASED ROBUSTNESS ON THE SHIFT FROM IMAGENET TO IMAGENET-C

To demonstrate the scalability of our approach, we perform experiments on ImageNet (Deng et al., 2009) and the recently-curated ImageNet-c dataset (Hendrycks & Dietterich, 2019). ImageNet-c contains images from the ImageNet test set that are corrupted according to artificial transformations, such as snow, rain, and fog, and are labeled from 1-5 depending on the severity of the corruption. For numerous challenging corruptions, we train models to map from the classes 0-9 of ImageNet to the corresponding classes of ImageNet-c. We then train all networks on classes 10-59 of ImageNet, and test on the corresponding classes for various subsets of ImageNet-c. Note that in this setting, the ImageNet classes used to train the model of natural variation are disjoint from those that are used to train the classifier, so many techniques, including most domain adaptation methods, do not apply; to offer a point of comparison, we include the accuracies of classifiers trained using AugMix, which is a recently proposed method that adds known transformations to the data (Hendrycks et al., 2019a).

## 5.3 ROBUSTNESS TO SIMULTANEOUS DISTRIBUTIONAL SHIFTS

In practice, it is common to encounter multiple simultaneous distributional shifts. For example, in image classification, there may be shifts in both brightness and contrast; yet while there may be examples corresponding to shifts in either brightness or contrast in the training data, there may not be any examples of both shifts occurring simultaneously. To address this robustness challenge,

Table 3: **Composing models of natural variation.** We consider shifts in two distinct and simultaneous sources of natural variation. To perform model-based training, we compose two models of natural variation trained separately on each of the two sources of natural variation.

| Dataset | Challenge 1 (dom. $A_1 \to$ dom. $B_1$) | Challenge 2 (dom. $A_2 \to$ dom. $B_2$) | Test acc. (top-1) | |
|---|---|---|---|---|
| | | | ERM | MDA |
| SVHN | Brightness (low$\to$high) | Contrast (low$\to$high) | 54.9 | **67.2** |
| ImageNet | IN-c brightness (low$\to$high) | IN-c contrast (high$\to$low) | 13.6 | **49.9** |
| ImageNet | IN-c brightness (low$\to$high) | IN-c snow (no$\to$yes) | 53.3 | **58.3** |
| ImageNet | IN-c brightness (low$\to$high) | IN-c fog (no$\to$yes) | 50.3 | **58.8** |
| ImageNet | IN-c contrast (high$\to$low) | IN-c fog (no$\to$yes) | 8.40 | **23.2** |

Table 4: **Transferability of model-based robustness.** In each experiment, we train a model of natural variation on a given training dataset $\mathcal{D}_1$. Then, we use this model to perform model-based training on a new dataset $\mathcal{D}_1$ entirely unseen during the training of the model.

| Training dataset $\mathcal{D}_1$ | Test dataset $\mathcal{D}_2$ | Challenge (dom. A$\to$dom. B) | Test accuracy (top-1) | | | | |
|---|---|---|---|---|---|---|---|
| | | | ERM | PGD | MRT | MDA | MAT |
| MNIST | Fashion-MNIST | Background color (blue$\to$red) | 69.3 | 67.7 | **81.4** | 80.1 | 76.1 |
| | Q-MNIST | | 87.0 | 79.9 | **98.0** | **98.0** | **98.0** |
| | E-MNIST | | 63.5 | 49.3 | **86.1** | 85.9 | 84.1 |
| | K-MNIST | | 47.9 | 47.7 | 89.1 | **89.3** | 86.8 |
| | USPS | | 89.9 | 87.4 | 93.3 | **93.4** | 91.9 |
| GTSRB | CURE | Brightness (high$\to$low) | 47.6 | 43.6 | **73.0** | 72.4 | 67.8 |
| ImageNet & ImageNet-c | CURE | Snow (no$\to$yes) | 52.0 | 53.0 | 59.4 | **62.2** | 59.4 |
| | | Brightness (low$\to$high) | 41.5 | 40.2 | 46.6 | 46.7 | **47.5** |

for each row of Table 3, we learn two models of natural variation $G_1$ and $G_2$ using unlabeled training data corresponding to two separate shifts, which map domains $A_1 \to B_1$ (e.g. low- to high-brightness) and $A_2 \to B_2$ (e.g. low- to high-contrast). We then compose these models to form a new model $G(x, \delta) = G_1(G_2(x, \delta), \delta)$ which can be used to provide robustness against both shifts simultaneously. We then train classifiers on labeled data from $A_1 \cap A_2$ and test on data from $B_1 \cap B_2$. To create the data from $B_1 \cap B_2$ for the ImageNet experiments, we apply pairs of transformations that were originally used to create the ImageNet-c datasets; more details are in Appendix D.

### 5.4 TRANSFERABILITY OF MODEL-BASED ROBUSTNESS

Because we learn models of natural variation offline before training a classifier, our paradigm can be applied to domains that are *entirely unseen* while training the model. In particular, we show that models can be reused on similar yet unseen datasets to provide robustness against a common source of natural variation. For example, one might have access to two domains corresponding to the shift from images of European street signs taken during the day to images taken at night. However, one might wish to provide robustness against the same shift from daytime to nighttime on a new dataset of American street signs without access to any nighttime images in this new dataset. Whereas many techniques, including most domain adaptation methods, do not apply in this scenario, in the MBRDL paradigm, we can simply learn a model corresponding to the changes in lighting for the European street signs and then apply this model to the dataset of the American signs. Table 4 shows several experiments of this stripe in which a model $G$ is learned on one dataset $\mathcal{D}_1$ and then applied on another $\mathcal{D}_2$; we improve robustness on unseen domains by up to 40 percentage points.

### 5.5 MODEL-BASED ROBUST DEEP LEARNING FOR UNSUPERVISED DOMAIN ADAPTATION

While our approach does not require labeled data from domain $B$, when such data is available, it is of interest to evaluate how our approach compares to relevant methods such as domain adaptation. In Table 5, for each shift from domain $A$ to $B$, we assume access to labeled data from domain $A$

Table 5: In each experiment, we assume access to unlabeled data from domain $B$, which we use to train a model of natural variation. We compare to suitable baselines, including domain adaptation.

| Dataset | Challenge (dom. A→dom. B) | Test accuracy (top-1) | | | | | |
|---|---|---|---|---|---|---|---|
| | | ERM | PGD | ADDA | MRT | MDA | MAT |
| SVHN | Brightness (low→high) | 30.5 | 36.2 | 60.1 | **70.9** | 69.5 | 52.2 |
| SVHN | Contrast (low→high) | 55.9 | 57.9 | 54.6 | **74.3** | 74.1 | 55.2 |
| GTSRB | Brightness (low→high) | 40.3 | 34.7 | 27.6 | 50.4 | 48.3 | **64.8** |
| GTSRB | Contrast (low→high) | 44.5 | 41.9 | 14.7 | 68.4 | **69.4** | 55.1 |
| CURE | Snow (no→yes) | 52.0 | 53.0 | 16.1 | 74.0 | **74.5** | 72.3 |
| CURE | Haze (no→yes) | 57.2 | 50.9 | 49.2 | 72.5 | 70.0 | **74.6** |
| CURE | Rain (no→yes) | 62.6 | 62.3 | 16.5 | 75.2 | 73.7 | **75.3** |

and unlabeled data from domain $B$. In each row, we use unlabeled data from both domains to train a model of natural variation. We then train classifiers using our algorithms, as well with ERM and PGD, using data from domain $A$ and test on data from the test set for domain $B$. Furthermore, we compare to ADDA, which is a well-known domain adaptation method (Tzeng et al., 2017). In every scenario, our model-based algorithms significantly outperform the baselines, often by 10-20 percentage points. Note that while this is one of the most commonly studied settings in domain adaptation, it represents only one particular setting to which the MBRDL paradigm can be applied.

## 6 RELATED WORK

Aside from the algorithms we introduced in Section 4, we are not aware of any other algorithms that can be used to address out-of-distribution robustness across the diverse array of tasks presented in the previous section. However, several lines of research have sought to address this problem in constrained settings or under highly restrictive assumptions. In the domain adaptation literature, various methods have been proposed which rely on the restrictive assumption that unlabeled data corresponding to a fixed distributional shift is available during training (Tzeng et al., 2017; Ajakan et al., 2014; Ganin & Lempitsky, 2015). Unlike these approaches, our solution does not assume access to unlabeled data from a fixed shift and can be applied to datasets that are entirely unseen during training. Furthermore, several works have used generative models to create adversarial perturbations (Xiao et al., 2018a; Lee et al., 2017; Wang & Yu, 2019; Samangouei et al., 2018; Jalal et al., 2017) or perceptually-realistic images subject to relatively simple corruptions in specific application domains (Dunn et al., 2019; Song et al., 2018; Vandenhende et al., 2019; Arruda et al., 2019). On the other hand, our approach is broadly applicable to arbitrary and challenging sources of natural variation.

Two concurrent works formulate robust training procedures assuming that data is corrupted according to a fixed generative architecture. Gowal et al. (2020) exploit properties specific to the StyleGAN architecture to formulate a training algorithm that provides robustness against color-based shifts on MNIST and CelebA. In our work, we propose a more general framework and three novel robust training algorithms that can exploit any suitable generative network, and we show improvements on more challenging, naturally-occurring shifts across twelve distinct datasets. Wong & Kolter (2020) use conditional VAEs to learn perturbation sets corresponding to simple corruptions from pairs of images. In our framework we improve robustness against more challenging, natural shifts by learning from *unpaired* datasets and we do not rely on class-conditioning to generate realistic images.

## 7 CONCLUSION

In this paper, we proposed a novel *model-based robust training paradigm* for deep learning that provides robustness with respect to models of natural variation. Our notion of robustness offers a departure from adversarial training with respect to norm-bounded data perturbations. In our experiments, we show that our paradigm can provide significant out-of-distribution robustness on many challenging distributional shifts. Furthermore, our paradigm can provide robustness against multiple simultaneous distribution shifts and on domains that are entirely unseen while training the model, and shows significant out-of-distribution robustness as datasets become more challenging.

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
