# OpenReview forum: "Model-Based Robust Deep Learning: Generalizing to Natural, Out-of-Distribution Data"
_ICLR.cc/2021/Conference — Reject_

### Official Review · AnonReviewer1 · 2020-10-28
**Review of "Model-Based Robust Deep Learning: Generalizing to Natural, Out-of-Distribution Data"**

**Rating:** 5
**Confidence:** 3

**Review:**

# Summary
This paper proposes a “paradigm shift” for augmenting datasets when training CNN-based image classifiers. On one side, traditional augmentations include blur, Gaussian noise, color distortions. On the other side, methods like adversarial training consider augmentations under norm bounds in the image space. The proposed method, instead, uses models of natural variation to augment the images. Increased out-of-domain accuracy is shown on ImageNet-C and several slices of the CURE-TSR dataset.

Central question: How to shift from perturbation-based robustness analysis to model-based robustness analysis?

# Strong and Weak points

## Strong points

  * This paper introduces a “paradigm shift” from perturbation based to model-based augmentation, when training CNN based image classifiers. The existing perturbations only reach so far, and using models of natural variation provides a new avenue for augmenting data sets.
  * The method is presented in clear pseudo code, and detailed in the 36 page counting appendix.

## Weak points

  * The paper claims a “paradigm shift”, but the proposed method of augmenting data has been used since 2012,  [1], and is used ever since [2]. Tables 2 and 3 provide key results for this method, but both were obtained with Model-based Data Augmentation, which is the common method before the "paradigm shift".
  * Results outlined in Section 5.2 and Table 2 concerns scenarios where the test-time perturbation is known at training time. Table 2 then compares MDA to ERM using this a-priori knowledge, and shows improved accuracy. However, the comparison is unbalanced, as ERM does not benefit from the prior knowledge. If Table 2 would be about comparing methods under the assumption that the test-time perturbation would be known at training time, then the ERM should be trained with the same data augmentation.
  * The paper misses comparisons against previous published work. Although Table 2 and 4 compare against another method, these results were not obtained by the respective authors, thus introducing a bias. The related work, Section 6, cites as many as 9 papers that addressed the same problem as this paper. The argument for the proposed method could be stronger when comparing to results from any of these papers.
Moreover, Table 3 reports results on ImageNet-C, which multiple other research papers have published results on. See for example, [3] and [4]. Likewise, Table 4 reports results on MNIST-C, which have been reported in [5] and [6]. Omitting those comparisons hampers the argument for the proposed method.

[1] Krizhevsky, 2012
[2] Szegedy, 2015
[3] https://paperswithcode.com/sota/domain-generalization-on-imagenet-c
[4] https://arxiv.org/pdf/1903.12261.pdf
[5] https://proceedings.icml.cc/paper/2020/file/20546457187cf3d52ea86538403e47cc-Paper.pdf
[6] https://arxiv.org/pdf/1906.02337.pdf

# Statement

Recommendation: Reject
Reason:
Although this paper provides four tables of extensive evaluation of the method, none of the numbers were obtained from previous literature. The related work section cites 9 papers on the same problem, but none were compared with in the results section. (See concrete details above)

# Clarifying questions

  * How to quantify the increase in computational complexity for the inner maximization in Equation (2)? As I understand, even evaluating one point requires a forward pass of the variation-model and the classifier-model. In case of iterative optimization, this would also require two backward passes(see line 5 in Algorithm (1)).
If the proposed method requires significantly more compute, the comparison should be made with a method that has access to the same amount of compute (for example a larger model, or longer trained model).
  * Line 6 of Algorithm (1). I miss the definition of set $\Delta$ and the projection operator in the text nor the appendix. Please explain?
  * Section 5.2 speaks of ImageNet “classes”. I am confused if these “classes” are referring to the ImageNet classes (1000 in total) or the severity levels in ImageNet-C (75 in total). Could you please clarify?
  Moreover, this argument implicitly assumes that the test-time perturbation is known at training-time. Could you name a scenario where this assumption is justified?
  * To what extent is the accuracy on the clean data preserved by the proposed method? Table 3, for example, shows an increase in top-1 accuracy on ImageNet data under perturbation. However, I wonder if this model preserves accuracy on the clean data. We know from existing literature [1] that a trade off exists for adversarial training between accuracy on perturbed and clean data. Providing the clean accuracy would provide evidence that this method compares favourably on this trade off.

[1] Zhang, Hongyang, et al. "Theoretically principled trade-off between robustness and accuracy." arXiv preprint arXiv:1901.08573 (2019).

# Minor feedback

This minor feedback is not part of the assessment.

  * “arguably more common” -> “arguably more important”
  * “x with a corresponding label y” -> “x with a corresponding prediction y”
  * Abbreviation “MBRDL” appears in Section 5 without any explanation or definition.
  * Abbreviation “CURE-TSR” appears in Section 5.0 without any explanation or definition.
  * The citation for CURE in Section 5.1 is wrong. The bibliography refers to “Temel, Dogancan, Min-Hung Chen, and Ghassan AlRegib. "Traffic sign detection under challenging conditions: A deeper look into performance variations and spectral characteristics." IEEE Transactions on Intelligent Transportation Systems (2019).”, but the CURE TSR dataset was published in “Temel, Dogancan, et al. "CURE-TSR: Challenging unreal and real environments for traffic sign recognition." arXiv preprint arXiv:1712.02463 (2017).”
  * Caption of Table 4: “ Then, we use this model to perform model-based training on a new dataset D1” -> “ Then, we use this model to perform model-based testing on a new dataset D2”?

---

> ### Author Response · Authors · 2020-11-17
> **Models of natural variation are **learned** from **unlabeled** data, which is quite different from plain data augmentation**
>
> Review response:
>
> **Data augmentation.**  While data augmentation has been used in many settings, the MDA algorithm uses two fundamentally different mechanisms from the papers in your review ([1,2]).  (1) First, we seek to **learn the underlying corruption or shift in the data distribution via models of natural variation**.  This is quite different from [1,2], in which pre-defined transformations are used for data augmentation.  This allows us to augment data corresponding to transformation for which there is no simple mathematical form, such as changes the weather conditions in images.  (2) Second, as our models of natural variation allow us to **produce multimodal output distributions over corrupted images by varying the nuisance parameter $\delta$**.  Thus, in each iteration of MDA, we continually look at new $\delta$s to generate images with different levels of natural variation.  This allows us, for example, to **continuously vary the amount of snow or rain added to images**, which contrasts prominently from [1,2].  Indeed, both of these features (learning corruptions from data and using multimodal and adaptive data augmentation using learned models of natural variation) are highly novel, and as far as we are aware, no other works (other than two concurrent papers [3,4], which we cite) have done anything similar.
>
> **On fair evaluation.**  We note that in Table 2 of Section 5.2, we have assumed access only to unlabeled data corresponding to a fixed transformation.  Thus, **it is not clear how one would use these unlabeled samples with ERM**, given that it is not possible to evaluate the loss function without labels.  We note that in all of our results, our algorithms never benefit from additional augmented labeled samples.  In fact, in Table 1, we show that even if ERM and PGD are given additional augmented labeled samples that MRT does not have access to, MRT still significantly outperforms these baselines.  While combining semi-supervision to use this unlabeled data with ERM is an interesting future direction, we believe it is beyond the scope of this work.
>
> **On comparisons to previous work.**
>
> In Table 2, we used the author’s implementation of AugMix, with the same settings and architecture as were used in the AugMix paper.  As the authors did not perform experiments in the setting of Table 2, it was impossible to obtain results on these experiments from these authors; we believe that using their code and their hyperparameters seems the most reasonable way to facilitate a fair comparison with AugMix.
>
> In Table 4, the original PGD paper does not perform experiments on any of the datasets we use, so it would be impossible to obtain results from these datasets from the original work. We provide our implementation of PGD in the code in the supplemental material.  In particular, we used $\epsilon = 8/255$, $\alpha=0.01$, and $n = 20$ steps of gradient ascent per batch; if the author feels that our implementation of PGD is unfair, we would be happy to discuss further.
>
> Contrary to the point raised in the review, **we do not report any results on MNIST-C.**  In Table 4, we use colorized versions of MNIST, Q-MNIST, E-MNIST, Fashion-MNIST, K-MNIST, and USPS, none of which overlap with the corruptions used in this MNIST-C dataset.  (We have provided samples from each of the colorized datasets used in Table 6 of the appendix.)  We would be happy to perform new experiments on MNIST-C if the reviewer would find this informative.
>
> **Related methods in our bibliography.**  In the related works section, most of the methods (other than the two concurrent works [3,4]) address the problem of creating Euclidean norm-bounded perturbations using generative models.  These works are related to our work in that they seek to use generative models in robustness applications, but they do so toward improving robustness against perturbations, which is an entirely different problem from the one we consider in this paper.  Moreover, we feel that by providing comparisons with PGD, which is known to be one of the strongest algorithms in this setting, we have shown that perturbation-based robustness does not improve natural, out-of-distribution robustness.  Thus we did not compare to additional perturbation-based algorithms.
>
> On the other hand, works such as [5] are only applicable in very specific settings, such as for shifts from day to night in images.  Finally, for several very recent methods such as  [6], there is not yet a public implementation available; when such an implementation becomes available, we would be happy to compare to this method.  We ask the reviewer: of the works that we cited (and/or those that we may have missed in our related work section), are there any in particular that the reviewer feels we should compare to?
>
> [1] Krizhevsky, 2012
> [2] Szegedy, 2015
> [3] Wong, Eric, and J. Zico Kolter, 2020
> [4] Gowal, Sven, et al, 2020
> [5] Arruda, Vinicius F., et al, 2019
> [6]Vandenhende, Simon, et al, 2019

---

> > ### Author Response · Authors · 2020-11-17
> > **Response to clarifying questions: computational complexity of MAT, definition of the nuisance space $\Delta$, a plausible real-world example of our scenario**
> >
> > Thank you for your conceptual questions at the end of your review.  We provide answers to these questions in this comment:
> >
> > **Computation complexity.**  The gradient-based scheme of Algorithm 1 does rely on gradient computation for both $G$ and $f$.  However, as the weights of $G$ are fixed while training the classifier, we do not perform an update step for $G$; only the weights of $f$ are updated.  The cost of the additional gradient computation for MAT at each step is not immense; however, this step does add to the complexity.  On the other hand, a MRT and MDA only require forward passes through $G$, and thus the computational complexity is similar to that of data augmentation.  In practice, this resulted in MAT talking approximately twice as long as MDA and MRT.
> >
> > **On the definition of $\Delta$ (nuisance space) and $\Pi$ (projection operator).**  The definition of $\Delta$ is discussed in detail in Appendix C.  In particular, notice that in Appendix C.2, we define $\Delta$ as the set of $\delta \in \mathbb{R}^q$ such that $-1 \preceq \delta \preceq 1$ where $q = 2$ or $q = 8$ depending on the dataset.  This particular choice of $\Delta$ is not essential, and we plan to investigate other choices in future work.  We use $\Pi$ to project to this set in the experiments.  In this way, the reviewer can think of the projection of a vector $\delta$ on $\Delta$ as clipping $\delta_j$ to fit in the range $-1 \leq \delta_j \leq 1$ for each $j = 1, \dots, q$.
> >
> > **On the definition of “classes” in Table 2.**  By classes, we mean the (total 1000) classes of the images in the original ImageNet dataset.  In these experiments, we allow the model of natural variation access to unlabeled corrupted images from the first 10 classes (classes 0-9 out of 1000) of ImageNet and a fixed subset of ImageNet-C, and then we train classifiers using this model on images from 50 different classes (classes 10-59 out of 1000) of ImageNet, and test on these same 50 classes (classes 10-59 out of 1000) on a fixed subset of ImageNet-C.
> >
> > Here is an example of a setting where this would be plausible.  Let’s say I’m an autonomous car driving around in Germany.  There is a big snowstorm coming, and to ensure safe driving behavior, I want my street sign classification algorithm to be robust against shifting snowy weather conditions.  To this end, I decide to train a model of natural variation to map from images of German street signs in sunlight to images of German signs in snowy weather.  Unfortunately, shortly before the snowstorm hits, I cross the border from Germany into the Netherlands (assuming that there are no pandemic-related restrictions).  I’ve driven in the Netherlands before, but I have only ever driven through the Netherlands in the summer, and thus I’ve never encountered snow there.  Moreover, the street signs are completely different in the Netherlands!  Not only are they differently shaped, but also, all of the words are in Dutch.  Now despite the fact that I’ve only ever encountered sunny street signs in the Netherlands, I can improve my robustness against the imminent snowstorm by using my pretrained model of natural variation, which was trained on a dataset containing only German street signs.
> >
> > This highlights a key feature of our framework: **models of natural variation can be trained on images from one distribution or dataset of images (in this example, German street signs) and then be applied to a new distribution (in this example, Dutch street signs)**.   Further, the fact that real-world examples exist for our problem is crucial to our motivation.  Indeed, **while it has been shown that small-perturbation-based adversarial examples can arise due to malicious tampering with the data, changes due to natural variation such as changes in weather conditions are much more likely to occur in real-world, dynamic environments.  This fact demonstrates the necessity for algorithms such as MAT, MDA, and MRT that can provide robustness against natural, out-of-distribution shifts in data.**
> >
> > **On clean accuracy.**  Throughout the experiments, we did not observe notable drops in clean accuracy between ERM and model-based training.  In particular, the difference on clean vs. robust accuracy on ImageNet/ImageNet-C was consistently within 5 percentage points between ERM and MDA.  We plan to devote a full paper to studying this trade-off in depth, as has been done for the problem of adversarial robustness [1].
> >
> > [1] Tsipras, Dimitris, et al. "Robustness may be at odds with accuracy." arXiv (2018).

---

### Official Review · AnonReviewer2 · 2020-10-28
**Extends adversarial learning to natural variations using GANs, but appears to have somewhat limited contribution**

**Rating:** 5
**Confidence:** 4

**Review:**

The paper extends current adversarial learning approaches beyond imperceptible L_p norm perturbations. The proposed approach can handle many models of natural variation, such as a change in brightness. The main idea behind the approach is to use unsupervised approaches such as GANs to model the natural variation. Given this model of natural variation, the paper replaces the adversarial learning objective of finding the worst example in an L_p norm ball around a data point to finding the worst example based on the model of natural variation. This is expensive, so the paper also proposes more computationally efficient approaches based on data augmentation. The experimental results demonstrate that the proposed approach performs well on a variety of tasks.

Strengths:

1. The question of training robust neural networks is clearly timely and well-motivated. A valid raised criticism of the current adversarial examples literature is that it is too tied to imperceptible or L_p norm perturbations, and this paper tries to go beyond them.
2. There is a bit of a lack of relevant methods to compare against, but the experiments still show that the proposed approach does well. I also appreciated the comparisons to domain adaption, and the multiple experimental setups generally.

Weaknesses:

1. Though the paper suggests that the approach is a “paradigm shift” in robust deep learning, it appears to be more or less an extension of the current adversarial examples literature. The adversarial training approach has been quite successful in defending against a fixed adversary model, from various L_p norms to geometric transformations. This paper is essentially defining a new adversary model, given by natural variations. In fact it is simpler than this, since the goal is to only do well on "random variations" in the predefined test set, i.e. there is no adversary which will try and attack the network by using the class of variations. There is a novelty here though that the variation model is learnt rather than pre-specified. But given that learning transformations such as change in brightness etc. is quite straightforward for unsupervised models such as GANs, this is not too much to ask for. Therefore, I find it a bit unsurprising that the proposed method works well, given the success of adversarial training and GANs.
2. Related to the above, in my view one main drawback of the current adversarial learning literature is that it is not difficult to get robustness to one pre-specified model of perturbation/variation, but in reality the space of possible variations is quite large. It is good that the proposed approach does seem to work on two simultaneous shifts, but the approach still seems restricted and not a substantial change in the current understanding. Note that some of the compared approaches are different in this regard: AugMix does not tailor the model to a particular variation (it explicitly excludes operations which overlap with ImageNet-C corruptions) and domain adaptation can handle quite large changes in the data distribution (which are not just transformations of the original data points).
3. Perhaps less importantly, in the experiments the proposed MDA approach based on data augmentation does almost as well as the other ones, and data augmentation using unsupervised models is already studied in the literature.

Overall, this is not a bad paper and I am not completely opposed to acceptance, but I am not sure I can argue for it.

Other less important comments:

1. Since there is a lack of baselines to compare against, the paper could benefit from a few ablation studies. For example, is there any difference on using other unsupervised approaches?
2. I think it would be better to have a concise list of a few major contributions in Section 1, rather than the current rather long list.
3. It would be nice to have representative images of the models of variation, perhaps in an appendix.

------Updates after author response------

I thank the authors for the response and it helps clarify some points. However, I am still not unconvinced that the paper is a significant departure from current work on adversarial robustness (see weakness 1 and 2 above). I think it would be much more interesting if the approach could yield robustness against a wider/different class of shifts compared to what it was adversarially trained against. Therefore, I am keeping my score at 5 and will not advocate for acceptance, though I am not completely opposed to it.

---

> ### Author Response · Authors · 2020-11-15
> **Enumerating the challenges in learning suitable models of natural variation and clarifying why our approach is much more sophisticated than plain data augmentation**
>
> Generally speaking, we agree with the reviewer that this is a very natural way of formulating the problem of improving robustness against natural, out-of-distribution shifts in data.  To the reviewer’s point, some corruptions such as brightness can be captured by various generative models.  However, capturing the multimodal nature of brightness, e.g. with different lighting colors as shown in Figure 7(d), is more challenging.  Still more challenging is accurately capturing phenomena such as rain or snow, which do not uniformly impact all pixels in an image, as opposed to changes in brightness or contrast.  **Our results on snow, rain, fog, and frost, in Tables 1, 2, and 3 and in particular the results that combine multiple sources of variation (see Table 3 and Figures 19-21 in the appendix) are quite challenging to capture.**  Thus, in many cases, the procedure we use to accurately capture natural corruptions in models of natural variation is highly nontrivial. **We have included many images in the appendix that illustrate models of natural variation (see Figures 5-17).**
>
> **AugMix.** We agree with the reviewer that AugMix is designed to solve a different problem from the setting in which we tested it.  Indeed, **as the reviewer points out, in our setting, there are a “lack of relevant methods to compare against.”  In our view, this is clear evidence of the novelty of our problem and our approach.**  However, we believe that of the very few methods that have been proposed toward improving more natural notions of robustness, AugMix is the most natural point of comparison, which is why it was included in this study.  Moreover, we note that AugMix uses a predefined set of transformations to perform data augmentation, and was shown to improve the mean corruption error on ImageNet-C and CIFAR-C [3].  To this end, while it has been shown that AugMix improves robustness on these datasets, it is unclear how to interpret why this method works, and whether it will generalize to meaningful kinds of corruptions outside of the “-C” datasets.  On the other hand, we emphasize that our approach differs from AugMix in that we seek to learn corruptions rather than applying pre-defined transformations.  **This feature of our approach allows us to apply different classes of problems, such as problems for which the corruptions used in AugMix (e.g. rotation, scaling, etc.) would not improve robustness.**
>
> **Domain adaptation.**  We also agree that domain adaptation can handle different settings, and in particular settings in which “are not just transformations of the original data points”.  However, as we are taking a robustness perspective in this paper, we are generally only interested in transformations that preserve the label of the original data.  Thus, in this setting, the main difference between our approach and the setting of unsupervised domain adaptation (UDA) is that in UDA, one generally assumes that unlabeled data from the target domain is available at training time.  In our work, we do not rely on this assumption (see, e.g., Table 4).  **Relaxing this assumption allows us to solve problems that domain adaptation cannot handle, such as scenarios in which no data from the target domain is available at training time.**  We include comparisons to UDA to demonstrate that our method outperforms UDA techniques in this setting, with the understanding that while this is one problem that the model-based robust deep learning framework is applicable to, there are also many other settings outside of domain adaptation in which we improve robustness.
>
> **Data augmentation.**  We agree with the review that data augmentation with unsupervised models has been done before (see [1,2]).  However, our approach, and in particular our MDA algorithm, differs fundamentally from these works.  Past work has sought to learn deterministic, 1-1 mappings between images in one domain to images in another [1,2].  This can be seen as deterministically mapping a dataset to a new domain and then training on the mapped data.  **However, in our approach, we vary the nuisance parameter to produce multimodal (e.g. one-to-many) mappings that map an input image to an output distribution of images with varying natural conditions.  Thus at each training iteration of MDA, we continually and adaptively look at new nuisance parameters to generate images with different levels of natural variation.**  Thus rather than simply using an unsupervised network to map one dataset to a new copy of that dataset, we continually generate new images to augment, which separates our algorithms from past work.
>
> [1] Arruda, Vinicius F., et al. "Cross-domain car detection using unsupervised image-to-image translation: From day to night." 2019.
> [2] Song, Yang, et al. "Constructing unrestricted adversarial examples with generative models." NIPS 2018.
> [3] Hendrycks, Dan, et al. "Augmix: A simple data processing method to improve robustness and uncertainty." arXiv (2019).

---

### Official Review · AnonReviewer3 · 2020-11-05
**Official Blind Review #3**

**Rating:** 5
**Confidence:** 3

**Review:**

This paper proposes a model-based framework for improving the robustness of image classifiers to average-case corruptions of varying severity. The proposed framework can be thought of as adversarial training where the perturbation is replaced by a function that transforms the image according to a specific corruption. A nuisance parameter controls the instantiation and severity of the corruption that is applied to the input. The paper compares baselines to different versions of this general model-based framework with experiments on several datasets.

Pros:
- Focusing on average-case corruptions is an important and underexplored problem
- Using image translation to learn the corruptions is an interesting proposal

Cons:
- There isn't much novelty in the proposed technique. Adversarial training with 'natural' corruptions has been done several times before.
- There isn't any discussion of how the nuisance parameter is used. I suspect it isn't used for ImageNet-C experiments, though I may be wrong. In this case, the method becomes data augmentation with image translation, which is not very novel either.
- It sounds like the ImageNet-C experiments use the corruptions to train the image translation network. Even though these experiments use held-out images, the corruption types themselves are not being held out, which goes against the recommended methodology for that dataset.


Typo: "ImageNet-c" should be "ImageNet-C"

___________________________________________________________________________________

Update after author feedback:

I thank the authors for improving my understanding of the paper. I feel better about it after reading it again. One of the most interesting findings--that a translation network trained on one domain/set of classes generalizes to another--needs more discussion. Using this phenomenon to improve robustness is a good idea, and the MRT/MDA/MAT methods explored in this work are nice choices for this investigation. However, I agree with the other reviewers that it would be nice to include more baselines from other work where appropriate.

The results in Table 1 are also very interesting and deserve more discussion--perhaps an analysis of whether a translation network trained on weak corruptions can generalize to create something akin to the ground-truth stronger corruptions, as the results in the table for MRT imply.

Overall, I think this paper has some strong experiments and investigates a good idea, but the claims of a "paradigm shift" are overly grandiose, and some of the most interesting experiments could use more analysis. Additionally, the writing clarity and presentation could be improved. My initial understanding of the paper was flawed in some places, so I'll raise my score to 5.

---

> ### Author Response · Authors · 2020-11-15
> **Our multimodal, optimization-based approach is more adaptive and refined than "image translation with data augmentation"**
>
> **The role of the nuisance parameter.**  **The nuisance parameter is the central mechanism by which we can vary images with respect to natural conditions.**  As shown in Figures 2(b), 5(d)-17(d), the multimodal distribution created by varying the nuisance parameter for learned models of natural variation engenders images that are perceptually and semantically very diverse.  For example, in Figure 7(d) of the appendix, we show that by varying the nuisance parameter $\delta$, we can produce images that contain the same bird as the original ImageNet image but that have very different levels of brightness.  That is, one value of $\delta$ corresponds to an image in a bright yellow light, whereas another value of $\delta$ corresponds to blue light being shined on the image, and so on.
>
> **A fundamental and highly novel idea in our work is to leverage the multimodal distributions induced by varying the nuisance parameter to train classifiers to be robust against challenging distributional shifts in data.**  In particular, observe that in equation (2), the nuisance parameter $\delta$ is the optimization variable of the inner maximization problem.  The goal of this inner maximization is to search over the space of nuisance parameters to find a nuisance parameter $\delta^\star$ that maximizes the loss $\ell(G(x,\delta^\star), y; w)$.  This means that we are searching for a nuisance parameter $\delta^\star$ that creates an image $G(x,\delta^\star)$ that is difficult for the classifier to correctly classify.  To this end, in each of our algorithms, we seek to (approximately) solve this inner maximization problem by either sampling different nuisance parameters $\delta$ or by performing gradient ascent in the nuisance parameter $\delta.$
>
> Thus, in all of experiments including those on ImageNet-C, we use the nuisance parameter to search for challenging natural conditions.  One subtlety here is that if a model of natural variation represented a deterministic, 1-1 mapping (which, in our paper, it does not) from images in one domain to images in another, then the current work would be more similar to past work [1,2], which as the reviewer points out constitute data augmentation with image translation.  **However, due to the ability of models of natural variation to produce multimodal output distributions, our algorithms use feedback from the training loop to continually and adaptively augment the training data with challenging examples subject to natural variation, in much the same way as PGD finds challenging examples to improve robustness against small perturbations in the adversarial robustness setting.**  This feature of our approach is highly novel, and as far as we are aware, no other works (other than two concurrent papers [3,4], which we cite) have done anything similar.
>
> Following this discussion, if the role or novelty of the nuisance parameter is still unclear, we would be happy to discuss the role of the nuisance parameter further.
>
> **The recommended usage of ImageNet-C.**  While we have not used the ImageNet-C dataset in the way that the authors originally specified [5], we see no issue with our usage.  Indeed, we are looking at the dataset in an entirely different way, wherein we train models of natural variation on one subset of classes and then train and test classifiers on another subset.  **This form of experiment introduces a new robustness challenge, in which we only see the corruption on a different distribution of images.  Thus, we believe that this new way of using ImageNet-C is actually a strength of our approach that will open up future avenues for research, rather than a weakness.**  If the reviewer has a specific concern about our usage of ImageNet-C, we would be happy to discuss further.
>
> **On novelty of adversarial training for natural corruptions.**  We are not aware of any work that uses multimodal image-to-image translation to formulate a min-max optimization problem to improve robustness against natural corruptions (aside from the concurrent works of [3,4]).  We emphasize that the multimodal nature of models of natural variation is crucial to our formulation; this is what allows us to look for challenging corruptions.  If the reviewer could provide any references that they believe are similar in this regard to our work, we would be more than happy to discuss these papers further.
>
> [1] Arruda, Vinicius F., et al. "Cross-domain car detection using unsupervised image-to-image translation: From day to night," 2019.
> [2] Song, Yang, et al. "Constructing unrestricted adversarial examples with generative models." NIPS, 2018.
> [3] Wong, Eric, and J. Zico Kolter. "Learning perturbation sets for robust machine learning." arXiv (2020).
> [4] Gowal, Sven, et al. "Achieving robustness in the wild via adversarial mixing with disentangled representations." CVPR, 2020.
> [5] Hendrycks, Dan, and Thomas Dietterich. "Benchmarking neural network robustness to common corruptions and perturbations." arXiv (2019).

---

### Official Review · AnonReviewer5 · 2020-11-06
**Unclear novelty and very similar to a broken robustness method ( defense )**

**Rating:** 5
**Confidence:** 2

**Review:**

The paper looks at the idea of building models of natural variation of an input and then using these models to develop robust training algorithms that are less susceptible to outliers ( testable for worst-cases via adversarial attacks ).

Strength:

+ The paper addresses a very important topic of adversarial robustness of DNN models and is accompanied by diligent evaluation over different datasets.

Weakness:

- The paper adopts an approach which is very reminiscent of Ajil Jalal, Andrew Ilyas, Constantinos Daskalakis, and Alexandros G Dimakis. The robust manifold defense: Adversarial training using generative models. arXiv preprint arXiv:1712.09196, 2017. The key difference seems to be adoption of the idea of using auxiliary transformations (called natural perturbations) which is also very well-studied in literature, for e.g. see https://openaccess.thecvf.com/content_CVPR_2020/papers/Zhang_Auxiliary_Training_Towards_Accurate_and_Robust_Models_CVPR_2020_paper.pdf  So, the approach presented here is quite incremental for a premier venue such as ICLR.

- The attack in Jalal et. al. paper can be launched against this approach once the attacker also access to this generative/natural model (which would be easy to build for an attacker)

- The reviewer will strongly recommend reviewing the advices in https://arxiv.org/abs/1902.06705 for writing papers on defense and how to self-evaluate its robustness by suitably designing the attacker. If the defense approach uses some background/auxiliary knowledge, one must consider the attacker with this knowledge if it is accessible to the attacker.

Questions to authors:

- Can authors explain why an attacker can't build similar natural model to defeat the proposed defense?
- Any clarification on incremental novelty from  Jalal et. al. and Zhang et. al. (and references therein) would be also useful.


After author's rebuttal:

"The other paper that the review points to ([4]) addresses a similar setting to our paper, but the approach is completely different. ... Moreover, [4] was published within three months of the ICLR submission deadline, meaning that it is essentially concurrent with our work."

Yes, the reviewer concurs that a work published so close to deadline should be treated as concurrent and will raise the score.

---

> ### Author Response · Authors · 2020-11-15
> **Fundamental misunderstanding about the topic and contribution of our work: this paper is about natural out-of-distribution robustness, not perturbation-based adversarial robustness**
>
> Unfortunately, we believe that this review has fundamentally misunderstood the point of our work.  In the summary and in the “Strengths” sections of this review, the reviewer writes that the point of our work is to “develop robust training algorithms that are less susceptible to adversarial attacks” and that our paper “addresses a very important topic of adversarial robustness.”  **We emphasize that this is not the goal of our work and that we do not study the setting of perturbation-based adversarial robustness at any point in this paper.**
>
> **In this paper, we study a completely different, highly novel problem, which concerns the robustness of deep learning to unbounded, natural, out-of-distribution shifts in data**, such as changes in lighting, weather, and background color in images.  We note that this problem setting arises due to the failure of existing adversarial training algorithms to provide meaningful levels of robustness to such unbounded shifts in the data distribution.  Indeed, the relatively poor performance of PGD compared to our algorithms across numerous out-of-distribution tasks in our results shows that adversarial training does not provide robustness to out-of-distribution shifts, and that **new methods are needed for improving robustness in this setting (for further justification of this fact, see [1,2,3])**.  To this end, we show that across 12 distinct datasets our algorithms improve out-of-distribution robustness significantly.
>
> **Comparisons to Jalal et al.**  Given this discussion, it should be clear that **the paper by Jalal et al. (which we cite on page 8) studies the fundamentally different problem of (perturbation-based) adversarial robustness, and thus our work should not be seen as “incremental” or even comparable to this paper.**  Moreover, not only does the paper by Jalal el al. study a different problem, but it also proposes a solution that is quite different from the methodologies we use.  In particular, Jalal et al. search for imperceptible adversarial examples by projecting images onto the span of a generator that maps random noise to an image manifold.  **In our setting, we search for naturally-varying images (e.g. with varying weather conditions) that are hard to classify by optimizing over the disentangled representations latent space of an image-to-image translation network.**  This image-to-image translation network maps an image to a multimodal distribution over output images.  By searching over this latent space, we find images with varying natural conditions that are difficult to classify.  As shown in Figures 2(b), 5(d)-17(d), the multimodal distributions produced by our models of natural variation contain images that are perceptually and semantically very different, which contrasts significantly with the generative models used in Jalal et al.  Further, our algorithms do not seek to project images onto the range of the generator; rather, we leverage disentangled representations to generate semantically meaningful variation, as opposed to norm-bounded perturbations.  **Lastly, as opposed to Jalal et al., there is no notion of a test-time adversary in our setting, as we are interesting in average-case performance.**
>
> The other paper that the review points to ([4]) addresses a similar setting to our paper, but the approach is completely different.  In [4], the authors rely on auxiliary classification networks, selective batch normalization, and weight decay to improve robustness.  **Our approach, in which we formulate a min-max optimization problem and use models of natural variation to describe specific shifts in the data distribution, is therefore fundamentally different from this approach.**  Moreover, [4] was published within three months of the ICLR submission deadline, meaning that it is essentially concurrent with our work.  That [4] was published so recently highlights the fact that the problem of addressing robustness to natural, non-adversarial shifts in the data distribution is an extremely timely and important problem.  Furthermore, because the authors have not yet released any implementation (as far as we can tell), we are not able to compare our results to [4] in the short window of the rebuttal period.
>
> Given this discussion, we ask the reviewer to reconsider their score, as we believe that there has been a fundamental misunderstanding about the nature and contribution of our paper and its relationship to the broader robustness literature.  If there are further questions, we are more than happy to continue this discussion.
>
> [1] Djolonga, et al. On robustness and transferability of convolutional neural networks. arXiv, 2020.
> [2] Taori, et al. Measuring robustness to natural distribution shifts in image classification. arXiv, 2020.
> [3] Hendrycks, et al. The many faces of robustness: A critical analysis of out-of-distribution generalization. arXiv, 2020.
> [4] Zhang, et al. Auxiliary Training: Towards Accurate and Robust Models.  CVPR (2020).

---

> > ### Comment · AnonReviewer5 · 2020-11-24
> > **Please look at the review again  ..**
> >
> > The strength point was identifying the importance of the broad area in absence of any other clear merit that the reviewer could identify. The reviewer has not misunderstood the paper. Any such perception is regrettable.
> >
> > As the summary statement clearly mentions "building models of natural variation of an input and then using these models to develop robust training algorithms". The use of adversarial attacks is a suggestion for the worst-case analysis but due to the confusion, the reviewer has lowered the confidence.
> >
> > Jalal et. al. considered statistical model of data instead of building explicit models as suggested in the paper. Their goal was robustness to adversarial perturbations but that is just an extreme worst-case analysis for any robustness model. As mentioned before, the attack in Jalal et. al. paper can be launched against this approach once the attacker also access to this generative/natural model (which would be easy to build for an attacker).  Let me try to explain again. There is no reason for adversarial attacks to be "perturbations" - recent attacks include introducing reflections - the key question in the review is what if we can train a Jalal et. al. approach to generate new OOD data which belong to similar natural outliers but fool the network.  Just because we have used adv methods to generate these outliers does not make them unnatural. These extreme case analysis would then demonstrate robustness or lack of robustness to natural outliers. Please look at https://arxiv.org/abs/1902.06705 in making claim of robustness.
> >
> > "The other paper that the review points to ([4]) addresses a similar setting to our paper, but the approach is completely different. ... Moreover, [4] was published within three months of the ICLR submission deadline, meaning that it is essentially concurrent with our work."
> >
> > Yes, the reviewer concurs that a work published so close to deadline should be treated as concurrent and will raise the score. But this should have helped authors realize that there is no "fundamental misunderstanding" of the paper.

---

> > > ### Author Response · Authors · 2020-11-24
> > > **The misunderstanding is that **there is no adversary** in our setting.  Our paper proposes a departure from adversarial robustness, which is a completely different problem.**
> > >
> > > We maintain that there is a fundamental misunderstanding here.  The crux of this confusion is that in our setting, **there is no adversary**.   Thus, your comment that says that we are
> > >
> > > > "building models of natural variation of an input and then using these models to develop robust training algorithms that are less susceptible to **adversarial attacks**"
> > >
> > > is incorrect, as our problem does not concern "adversarial attacks".  As shown throughout our experiments, we are interested in *average-case, out-of-distribution* robustness.  Note that this notion of robustness differs considerably from adversarial robustness, which considers *worst-case, in-distribution* robustness.
> > >
> > > Perhaps this has not been clearly conveyed in our writing, and we hope to revise the writing to better reflect this fact.  However, we hope that the reviewer will agree that there are many important notions of robustness that do not require an adversary.  For example, in an autonomous driving application, there is no malicious agent that changes the weather conditions to be as challenging as possible at test time.  Rather, weather conditions vary in natural, continuous ways.  Therefore, we feel that new notions of robustness that do not adhere to the "adversarial," worst-case mindset for such scenarios are needed.  Our paper is one of the first works to begin to address this gap.
> > >
> > > If the reviewer disagrees with this characterization of our work, or if the reviewer feels that notions of robustness outside of adversarial robustness are not important for the deep learning community, we would be happy to provide further motivation and/or to discuss this further.
> > >
> > > ___
> > >
> > > To this end, we find that the authors comment that there is an
> > >
> > > > "absence of any other clear merit that the reviewer could identify"
> > >
> > > misses the highly-novel notion of natural, out-of-distribution robustness that we are proposing.  Whereas a plethora of works have considered adversarial robustness, and as you point out a set of community guidelines for adversarial robustness has been established (https://arxiv.org/abs/1902.06705), there is not yet any kind of standard for the notion of robustness we introduce in this paper.
> > >
> > > ___
> > >
> > > Given this departure from adversarial robustness, the comment
> > >
> > > > "As mentioned before, the attack in Jalal et. al. paper can be launched against this approach once the attacker also access to this generative/natural model (which would be easy to build for an attacker)."
> > >
> > > is not applicable.  As there is no test-time adversary, this "attack" is not relevant in our setting.  Furthermore, the attack is only designed to generate images that are **imperceptible** from images in the true data distribution.  The reviewer mentions that
> > >
> > > > "the key question in the review is what if we can train a Jalal et. al. approach to generate new OOD data which belong to similar natural outliers but fool the network. "
> > >
> > > **but this was never done in Jalal et al.**  Indeed, as far as we are aware, **this idea has never been tried before.**  Thus we feel that the criticism that an attack that has never been tried before for **perceptable, out-of-distribution** shifts might decrease our accuracy is not supported by any evidence.

---

> > > > ### Comment · AnonReviewer5 · 2020-11-24
> > > > **Goal of adversarial attack over natural variations is to stress-test robustness to provide evidence of method's effectiveness**
> > > >
> > > > The reviewer went over the paper again and is not convinced about novelty or that the presented approach is not broken (would not generalize - as can be found through stress-testing with not the typical adversarial attacks but by finding manifold of natural variations - Jalal reference as a guide to learn manifold of variations - and using that to craft adversarial attacks as suggested above).  The evaluation of a scientific approach requires due diligence which is significantly lacking in the paper.
> > > >
> > > > Reviewer accepts that comparison with CPVR 2020  paper Zhang, et al. Auxiliary Training: Towards Accurate and Robust Models. CVPR (2020) can be excused given an almost concurrent publication. But the explanations provided by the authors in the response are sound and can be included in the final version. Reviewer would also suggest an empirical comparison for completeness if authors are willing to do so.
> > > >
> > > > But that still leaves a burden of demonstrating utility and effectiveness of the approach. The reviewer does not agree with authors in the claim that suggestions of deploying adversarial attacks that search over natural variations and find extreme cases to demonstrate robustness are completely off-topic and irrelevant.  The fact that the paper's goal is not to defend against adversarial attacks does not preclude using "not perturbations" but instead the suggested adversarial/worst-case generation of natural variations to test the presented approach. In the absence of such an analysis, the reviewer is not confident the presented approach has demonstrated its value.  The goal is not to demonstrate the proposed approach is robust to all worst-case perturbations but rather to show incremental improvement - this would establish that the presented approach is truly achieving generalized robustness.
> > > >
> > > > The misunderstanding of the authors originate from the use of the phrase "adversarial attack" - but these are just worst-case input generation methods and instead of searching over the image manifold as in Jalal paper, the proposed robustness method can be evaluated over the manifold of natural variations. Can the author's explain why such an approach would be an unfair way to evaluate the  claimed robustness?
> > > >
> > > > Based on the current draft, this paper is not yet ready to be accepted in a premier venue such as ICLR in the view of this reviewer. Having said that, the reviewer will not argue against accepting the paper. Consequently, the reviewer has lowered the confidence.

---

> > > > > ### Author Response · Authors · 2020-11-24
> > > > > **Further clarifications on Jalal et al., average-case vs. worst-case evaluation**
> > > > >
> > > > > Thanks for following up.  We appreciate your engagement with our paper.  Here are some further clarifications:
> > > > >
> > > > > > "The reviewer ... is not convinced about novelty or that the presented approach is not broken"
> > > > >
> > > > > In proposing this new robustness framework that does not rely on an adversary at test time, in this paper we define a new kind of robustness that does not rely on "attacks" or "defenses."  Note that this is totally different from adversarial robustness, and we encourage the reviewer to consider that attacks and defenses are just one way to think about robustness.
> > > > >
> > > > > Because we are interested in average-case performance, there is no notion of "breaking a defense" in our paper, because "breaking a defense" implies that our evaluation metric relies on an adversary that is able to find a worst-case example; clearly average-case performance does not rely on such an adversary.  Thus, in our framework, it doesn't make sense to "attack" or "defend" against any kind of attack; this is one reason why Jalal et al. is not applicable in our setting.
> > > > >
> > > > > ___
> > > > >
> > > > > > "...the [Jalal et al.] robustness method can be evaluated over the manifold of natural variations. Can the author's explain why such an approach would be an unfair way to evaluate the claimed robustness?
> > > > >
> > > > > In Jalal et al, the main idea is to learn a spanning network/generator $H: Z\to X$ where $Z$ is a low-dimensional noise space and $X$ is the data space.  Thus, $H$ can be used to map latent codes, which have no semantic meaning, to the original image manifold.  As the adversarial examples crafted in Jalal et al. are imperceptible, the authors consider small perturbations in the noise space to induce images on the range of $H$ that are imperceptible from one another.
> > > > >
> > > > > It is unclear how this would be extended to the case of natural variation.  The fundamental question in generating natural variation is: given an image $x$, how can I change the natural conditions in $x$ (e.g. add snow or rain).  To do this in the Jalal et al. framework, one issue is that for any input image $x$, one would need to be able to identify a direction in the latent space that changed natural conditions (such as adding snow).  It is unclear how this would be done, especially because $H$ is not trained to have any such interpretable directions in the latent space, and indeed Jalal et al do not follow this approach.  Indeed, achieving this would be a significant contribution and would likely be worthy of a paper in its own right, especially given that finding interpretable directions in the latent space of GANs is still an open problem (see, e.g. https://openreview.net/forum?id=HylsTT4FvB).
> > > > >
> > > > > Thus, the Jalal et al. framework is not easily adaptable to natural variation.  A more natural way to model natural variation is to learn a map that takes as input an image $x$ AND a noise vector $\delta$ (rather than only a noise vector) and outputs an image $x$ with different natural conditions.  In this way, such a function would be dependent on the original input image, and the space of $\delta$s would be explicitly trained to add natural conditions to that any input image.  Note that this circumvents the problem that one would have in Jalal et al of identifying a direction corresponding to a source of natural variation, since $\delta$ is designed to fulfill this role.  And indeed, this is exactly the approach we take in this paper by learning models of natural variation $G:X\times\Delta \to X$ where $\Delta$ is the space of interpretable $\delta$s.
> > > > > ___
> > > > >
> > > > > > "But the explanations provided by the authors in the response are sound and can be included in the final version."
> > > > >
> > > > > We will be happy to mention this work and provide a comparison along the lines of what was previously discussed in this thread in the final version of this paper.
> > > > >
> > > > > ___
> > > > >
> > > > > > "...deploying adversarial attacks that search over natural variations and find extreme cases to demonstrate robustness are completely off-topic and irrelevant"
> > > > >
> > > > > We do not think that these attacks are "off-topic"; they simply address a different problem.  The adversarial robustness literature was relevant in the identification of the problem itself, which is that many forms of robustness do not require "attacks" or "defenses."  An extreme case analysis may be interesting, but this is not the topic under consideration here.
> > > > >
> > > > > Here's another way to look at it.  When proposing a defense to adversarial examples, one would not fault a paper if it did not report the accuracy on randomly perturbed samples; the worst-case/adversarial accuracy is the relevant metric, not the average-case.  In the same way, in our setting, the average-case performance is the important metric, whereas the "worst-case" performance, which potentially interesting in other applications, is not the topic of this manuscript.

---

> > > > > > ### Author Response · Authors · 2020-11-24
> > > > > > **On incremental improvement and general robustness**
> > > > > >
> > > > > > > "demonstrate the proposed approach is robust to all worst-case perturbations but rather to show incremental improvement - this would establish that the presented approach is truly achieving generalized robustness."
> > > > > >
> > > > > > In this paper, we are not interested in incremental improvement over adversarial robustness methods.  Adversarial examples fall outside of the scope of natural variation given their imperceptible nature.  To be clear, worst-case natural variation is also outside the scope of our paper, as we are interested in average-case performance.
> > > > > >
> > > > > > Moreover, we are not looking for a method that solves all robustness challenges in ML or that is generally robust to BOTH adversarial examples and natural variation.  This paper only concerns robustness challenges that involve natural variation.  A general way of improving robustness against natural variation and adversarial examples is an interesting problem, but again, it is not the problem under consideration in this paper.

---

### Author Response · Authors · 2020-11-24
**Post-rebuttal message to the reviewers of this paper**

Dear reviewers,

We sincerely appreciate the time and effort you took to review our paper.

We would like to thank those reviewers that have raised their scores post-rebuttal.  As this is the last day in which discussion can take place between authors and reviewers, we would also greatly appreciate responses from Reviewers 2 & 3, as our rebuttal clarifies several important points brought up in their reviews.

To summarize, our paper addresses the important problem of robustness against natural, out-of-distribution shifts in data.  This is fundamentally different from adversarial robustness and plain data augmentation, and our results show remarkable levels of out-of-distribution robustness across 12 distinct datasets.  Given the highly novel problem we propose and our principled approach that seeks to learn underlying distributional shifts from data, we ask the reviewers to consider that this problem may have a significant impact on the way different forms of robustness are studied in deep learning moving forward, as the community shifts from adversarial, worst-case robustness to more natural notions of distributional, average-case robustness.

Many thanks,

The authors

---

### Decision · Program_Chairs · 2021-01-07
**Final Decision**

**Decision:**

Reject

**Comment:**

# Quality:
The technical contribution of the paper seems reasonable and there were only minor points being highlighted by the reviewers.

# Clarity:
The paper would benefit from being more polished. During the rebuttal, the authors suggested that several reviewers misunderstood the paper. This alone should encourage the authors to improve clarity.

# Originality:
Several reviewers presented concerns about the claims of the authors and the existence of connections to existing literature. Nonetheless, the proposed approach seems novel to the best of the reviewers and my knowledge.

# Significance of this work:
The topic of the manuscript is relevant and impactful. However, several reviewers suggested to include additional baselines in the experiments to validate the goodness of the proposed approach.

# Overall:
The paper presents an interesting idea, with a high potential impact. Despite the interesting topic and some interesting insights, all the reviewers agree that the manuscript is not ready for publication just yet. I want to encourage the authors to keep improve it and resubmit it at the next conference.